# Analytic neutrino oscillation probabilities

**Chee Sheng Fong**⋆

Centro de Ciências Naturais e Humanas, Universidade Federal do ABC,
09.210-170, Santo André, SP, Brazil

⋆ sheng.fong@ufabc.edu.br

## Abstract

In the work, we derive exact analytic expressions for $(3 + N)$-flavor neutrino oscillation probabilities in an arbitrary matter potential in term of matrix elements and eigenvalues of the Hamiltonian. With the analytic expressions, we demonstrate that nonunitary and nonstandard neutrino interaction scenarios are physically distinct: they satisfy different identities and can in principle be distinguished experimentally. The analytic expressions are implemented in a public code `NuProbe`, a tool for probing new physics through neutrino oscillations.

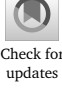

# 1 Introduction

Neutrino oscillation is an intriguing quantum mechanical phenomenon that provides one of the first definite evidences for physics beyond the Standard Model (SM). While the standard three-flavor neutrino oscillation phenomena with the SM matter potential is well-established and rather successful, the origin of neutrino mass, which necessitates new degrees of freedom beyond the SM, is still an open problem. With increasing precision in neutrino oscillation experiments, one might start to see deviations from the standard three-flavor neutrino oscillation paradigm. For instance, a deviation from unitary leptonic mixing matrix is expected to be measurable [1–18] if the origin of neutrino mass is of relatively low scale. Even more note worthily, if neutrinos are quasi-Dirac, three-flavor neutrino oscillation picture is no longer sufficient [19–23]. By studying neutrino oscillation in matter, one might also discover new neutrino interactions beyond that of the SM [24–26]. Motivated by this, in Section 2, we will review the neutrino oscillation formalism that generalizes the three-flavor oscillation paradigm to oscillations among $3+N$ flavor states in an arbitrary matter potential. While analytic formulas have been obtained for the standard three-flavor neutrino oscillation probability [27–33], in this work, we further derive simple analytic formulas to describe $(3+N)$-flavor neutrino oscillation in terms of Hamiltonian elements and its eigenvalues as was first done by Yasuda in ref. [34] in Section 3. We also clarify the distinction between nonunitary and NonStandard neutrino Interaction (NSI) scenarios.[1] In the former, Naumov-Harrison-Scott (NHS) identity is violated and unitary relations are replaced by new identities while in the latter, NHS is violated only if the matter potential is nondiagonal. In all cases, unitary relations remain exact if the evolution is unitary. Analytic results for $(3+1)$-flavor scenario were obtained previously in refs. [36–41], and in Appendix A, we present again these exact results in a general and compact form. In Appendix B, we describe the Python code `NuProbe` with in-built analytic solutions up to $(3+4)$-flavor neutrino oscillation system and illustrate example applications in nonunitary, NSI, and quasi-Dirac neutrino scenarios.

# 2 Review of $3+N$ neutrino oscillations

Let us consider the case where the neutrino flavor states $|\nu_\alpha\rangle$ are related to the mass eigenstates $|\nu_i\rangle$ through a unitary matrix $U$

$$|\nu_\alpha\rangle = \sum_i U^*_{\alpha i} |\nu_i\rangle \, , \tag{1}$$

where $\alpha = e, \mu, \tau, s_1.s_2, ..., s_N$ and $i = 1, 2, ..., 3+N$. Here $\nu_e$, $\nu_\mu$ and $\nu_\tau$ are the SM neutrino flavor states. The time-evolved state $|\nu_\alpha(t)\rangle$ with $|\nu_\alpha(0)\rangle = |\nu_\alpha\rangle$ is described by the Schrödinger equation

$$i\frac{d}{dt}|\nu_\alpha(t)\rangle = \mathcal{H}|\nu_\alpha(t)\rangle \, , \tag{2}$$

where we separate the Hamiltonian $\mathcal{H} = \mathcal{H}_0 + \mathcal{H}_I$ with $\mathcal{H}_0$ the free Hamiltonian

$$\mathcal{H}_0|\nu_i\rangle = E_i|\nu_i\rangle \, , \qquad E_i = \sqrt{|\vec{p}_i|^2 + m_i^2} \, , \tag{3}$$

and $\mathcal{H}_I$ the interaction Hamiltonian with matrix elements

$$\langle\nu_\beta|\mathcal{H}_I|\nu_\alpha\rangle = V_{\beta\alpha} \, . \tag{4}$$

---

[1]A study on how to identity new neutrino oscillation physics scenarios at DUNE experiment is presented in ref. [35].

New physics can enter through additional $N$ neutrinos which do not feel the weak interactions (the sterile neutrinos) and/or new interactions of the SM neutrinos and/or of the sterile neutrinos which enter in $V$.

Since we do not measure the propagation time of neutrinos but the distance $x$ traveled by them, we will trade $t = x$ assuming relativistic neutrinos. The amplitude of the transition $\nu_\alpha \to \nu_\beta$ at distance $x$ is $S_{\beta\alpha}(x) \equiv \langle \nu_\beta | \nu_\alpha(x)\rangle$ and the probability of neutrino starting from $|\nu_\alpha(0)\rangle = |\nu_\alpha\rangle$ and being detected as $|\nu_\beta\rangle$ at distance $x$ is

$$P_{\beta\alpha}(x) = \left| S_{\beta\alpha}(x) \right|^2 . \tag{5}$$

From eqs. (1)–(4), we can write the evolution equation of $S_{\beta\alpha}(x)$ as

$$i\frac{d}{dx}S_{\beta\alpha}(x) = \sum_\gamma \left[ \sum_i U_{\beta i} E_i U_{\gamma i}^* + V_{\beta\gamma} \right] S_{\gamma\alpha}(x) . \tag{6}$$

For relativistic neutrinos, we can approximate $E_i \simeq E + \frac{m_i^2}{2E}$ and dropping the constant term $E$ (which is an overall phase), we have, in matrix notation $i\, dS(x)/dx = HS(x)$ where $H \equiv U\Delta U^\dagger + V$ and

$$\Delta \equiv \frac{1}{2E}\mathrm{diag}\left(m_1^2, m_2^2, ..., m_{3+N}^2\right) = \mathrm{diag}(\Delta_1, \Delta_2, ..., \Delta_{3+N}) . \tag{7}$$

The formal solution is

$$S(x) = T\exp\left[-i\int_0^x dx' H\left(x'\right)\right], \tag{8}$$

where $T$ stands for "space ordering". If $H(x)$ is independent of $x$ e.g. in the vacuum or $V$ is independent of $x$, we have $S(x) = e^{-iHx}$.

It is convenient to work in the *vacuum mass basis*

$$\widetilde{S}(x) = U^\dagger S(x) U , \tag{9}$$

in which $i\, d\widetilde{S}(x)/dx = \widetilde{H}\widetilde{S}(x)$ where $\widetilde{H} = U^\dagger H U = \Delta + U^\dagger V U$ is the Hamiltonian in the vacuum mass basis. If $\widetilde{H}$ is independent of $x$ in the interval of interest $0 < x < x_1$, we can diagonalize $\widetilde{H}$ as follows

$$\widetilde{H} = X\hat{H}X^\dagger , \tag{10}$$

where $X$ is unitary and $\hat{H} = \mathrm{diag}(\lambda_1, \lambda_2, ..., \lambda_{3+N})$ is diagonal and real and hence $\widetilde{S}(x) = Xe^{-i\hat{H}x}X^\dagger$. From eq. (9), we have $S(x) = UXe^{-i\hat{H}x}(UX)^\dagger$ and from eq. (5), neutrino oscillation probability for $\nu_\alpha \to \nu_\beta$ is

$$P_{\beta\alpha}(x) = \left| \sum_{i,j,k} U_{\beta i} U_{\alpha j}^* X_{ik} X_{jk}^* e^{-i\lambda_k x} \right|^2 . \tag{11}$$

The oscillation probability $\overline{P}_{\beta\alpha}(x)$ for antineutrino $\overline{\nu}_\alpha \to \overline{\nu}_\beta$ is obtained by taking $U_{\alpha i} \to U_{\alpha i}^*$ and $V \to -V$ in eq. (6). So even if the $U$ is real (CP-conserving), in general $\overline{P}_{\beta\alpha}(x) \neq P_{\beta\alpha}(x)$ due to the potential consisting of only matter. Denoting $\widetilde{U} \equiv UX$, eq. (11) can also be written in a more familiar form

$$P_{\beta\alpha}(x) = \delta_{\alpha\beta} - 2\sum_{j\neq k}\mathrm{Re}\left(\widetilde{U}_{\beta j}\widetilde{U}_{\alpha j}^*\widetilde{U}_{\beta k}^*\widetilde{U}_{\alpha k}\right)\sin^2\frac{(\lambda_k - \lambda_j)x}{2}$$
$$- \sum_{j\neq k}\mathrm{Im}\left(\widetilde{U}_{\beta j}\widetilde{U}_{\alpha j}^*\widetilde{U}_{\beta k}^*\widetilde{U}_{\alpha k}\right)\sin\left[(\lambda_k - \lambda_j)x\right] . \tag{12}$$

For three-flavor scenario in vacuum $V = 0$ and $X = I_{3\times3}$, we recover the standard neutrino oscillation probability in the vacuum.

If $V$ is $x$-dependent, we can split $x$ into elements of $dx$, small enough that $V(x)$ is approximately constant and construct the full solution by matching the solutions between subsequent intervals. Considering $0 = x_0 < x_1 < x_2 < ...$ where $V(x)$ is equal to constant $V_a$ for each interval $x_{a-1} < x < x_a$, the full solution is

$$S = T\prod_{a=1} S^{(a)}, \tag{13}$$

where we have defined

$$S^{(a)} \equiv \left(UX^{(a)}\right)e^{-i\hat{H}^{(a)}x^{(a)}}\left(UX^{(a)}\right)^{\dagger}, \tag{14}$$

$$x^{(a)} \equiv \left[(x - x_{a-1})\,\theta\,(x_a - x) + (x_a - x_{a-1})\,\theta\,(x - x_a)\right]\theta\,(x - x_{a-1}), \tag{15}$$

with $\theta(x \geq 0) = 1$ and $\theta(x < 0) = 0$ and the space ordering of the matrix multiplication is such that the $a$ term is always to the left of $a-1$ term. $\hat{H}^{(a)} = \mathrm{diag}\left(\lambda_1^{(a)}, \lambda_2^{(a)}, ..., \lambda_{3+N}^{(a)}\right)$ and $X^{(a)}$ denote respectively the matrix of eigenvalues and unitary matrix which diagonalizes $\widetilde{H}^{(a)} = \Delta + U^{\dagger}V_a U$ as $\widetilde{H}^{(a)} = X^{(a)\dagger}\hat{H}^{(a)}X^{(a)}$ in the interval $x_{a-1} < x < x_a$. The neutrino oscillation probability can be calculated by substituting eq. (13) into eq. (5).

Notice that just like in eq. (11), for each layer, $X^{(a)}$ always appears in the combinations $X_{ik}^{(a)}X_{jk}^{(a)*}$ in the transition amplitude $S^{(a)}$ as follows

$$S_{\beta\alpha}^{(a)} = \sum_{i,j,k} U_{\beta i}U_{\alpha j}^* X_{ik}^{(a)}X_{jk}^{(a)*}e^{-i\lambda_k^{(a)}x^{(a)}}. \tag{16}$$

In the next section, we will derive the analytic solutions for $X_{ik}^{(a)}X_{jk}^{*(a)}$ which allow us to write down analytic expressions for $3 + N$ neutrino oscillation probability in an arbitrary matter potential.

## 3 Analytic solutions

We would like to solve $X_{ik}X_{jk}^*$ analytically in terms of the eigenvalues and the matrix elements of $\widetilde{H}$. Here we drop the superscript $(a)$ focusing on each interval where $V$ is constant and the solution for a generic $V(x)$ can be constructed as in eq. (13).

### 3.1 $(3 + N)$ – flavor scenario

Let us consider the general case with $3+N$ neutrino flavor states. We start by noticing that [34]

$$\sum_k X_{ik}X_{jk}^* = \delta_{ij},$$

$$\sum_k \lambda_k X_{ik}X_{jk}^* = (\widetilde{H})_{ij},$$

$$\sum_k \lambda_k^2 X_{ik}X_{jk}^* = (\widetilde{H}^2)_{ij}, \tag{17}$$

$$\vdots$$

$$\sum_k \lambda_k^{2+N}X_{ik}X_{jk}^* = (\widetilde{H}^{2+N})_{ij},$$

where the first equation follows from the unitarity of $X$ while the rest follow directly from eq. (10). So we have a set of linear equations in $X_{ik}X_{jk}^*$ where the coefficients form a Vandermonde matrix which can be readily inverted to give[2]

$$X_{ik}X_{jk}^* = \frac{\sum\limits_{p=0}^{2+N}(-1)^p(\widetilde{H}^p)_{ij}c_{2+N-p,k}}{Z_k}\,,\tag{18}$$

where we have defined

$$Z_k \equiv \prod_{p\neq k}(\lambda_p - \lambda_k)\,,\qquad c_{p,k}\equiv\sum_{\{q\neq r\neq\ldots\}\neq k}\underbrace{\lambda_q\lambda_r\ldots}_{p}\,,\tag{19}$$

with $(\widetilde{H}^0)_{ij}=\delta_{ij}$, $c_{0,k}=1$ and the sum in $c_{p,k}$ is over all possible unordered combinations of $p$ distinct eigenvalues $\lambda_q\lambda_r\ldots$ where none of them is equal to $\lambda_k$. With $3+N$ neutrino flavors, $c_{p,k}$ has altogether $\binom{2+N}{p}=\frac{(2+N)!}{p!(2+N-p)!}$ terms in the sum.[3] If we have $d+1$ degenerate eigenvalues $\lambda_l = \lambda_k$ for $l = k,\ldots,k+d$, then we only need to solve for the combination $\sum\limits_{l}X_{il}X_{jl}^*$ corresponding to $\lambda_k$. The rank of system of linear equations is reduced from $3+N$ to $3+N-d$ or effectively, we have a $(3+N-d)$-flavor scenario.

## 3.2 Three – flavor scenario

Three-flavor scenario is of great interest since we know that the SM comes in three weakly-interacting neutrinos and more importantly, one can probe new physics if $V$ is modified due to new physics interactions and/or nonunitarity in $U$ is induced due to the existence of sterile neutrinos. As shown in refs. [10, 13], if one can average out the fast oscillations involving sterile neutrinos which participate in neutrino oscillations, the leading term in the vacuum mass basis Hamiltonian is still given by $\widetilde{H}=\Delta+U^\dagger V U$ with a nonunitary $U$. The characteristic equation of $\widetilde{H}$ can be constructed using the Faddeev-LeVerrier algorithm

$$\lambda^3 - \mathcal{T}\lambda^2 + \mathcal{A}\lambda - \mathcal{D} = 0\,,\tag{20}$$

where we have defined

$$\mathcal{T}\equiv\mathrm{Tr}\widetilde{H}\,,\qquad\mathcal{D}\equiv\det\widetilde{H}\,,\qquad\mathcal{A}\equiv\frac{1}{2}\left(\mathcal{T}^2-\mathcal{T}_2\right)\,,\tag{21}$$

with

$$\mathcal{T}_p\equiv\mathrm{Tr}(\widetilde{H}^p)\,.\tag{22}$$

The three real eigenvalues of $\widetilde{H}$ can be obtained from the Cardano formulas

$$\lambda_{1,2}=\frac{\mathcal{T}}{3}-\frac{1}{3}\mathcal{F}\cos\mathcal{G}\mp\frac{1}{\sqrt{3}}\mathcal{F}\sin\mathcal{G}\,,\qquad\lambda_3=\frac{\mathcal{T}}{3}+\frac{2}{3}\mathcal{F}\cos\mathcal{G}\,,\tag{23}$$

where we have defined

$$\mathcal{F}\equiv\sqrt{\mathcal{T}^2-3\mathcal{A}}\,,\qquad\mathcal{G}\equiv\frac{1}{3}\arccos\left(\frac{2\mathcal{T}^3-9\mathcal{A}\mathcal{T}+27\mathcal{D}}{2\mathcal{F}^3}\right)\,.\tag{24}$$

---

[2]See the beautiful exposition on the identity between eigenvectors and eigenvalues in ref. [42].

[3]For instance, for $3+2$ neutrino flavors, we have

$$c_{2,3}=\lambda_1\lambda_2+\lambda_1\lambda_4+\lambda_1\lambda_5+\lambda_2\lambda_4+\lambda_2\lambda_5+\lambda_4\lambda_5\,.$$

From eq. (18), the mixing elements are[4]

$$X_{i1}X_{j1}^* = \frac{\delta_{ij}\lambda_2\lambda_3 - (\widetilde{H})_{ij}(\lambda_2+\lambda_3) + (\widetilde{H}^2)_{ij}}{(\lambda_2-\lambda_1)(\lambda_3-\lambda_1)}\,, \tag{25a}$$

$$X_{i2}X_{j2}^* = \frac{\delta_{ij}\lambda_1\lambda_3 - (\widetilde{H})_{ij}(\lambda_1+\lambda_3) + (\widetilde{H}^2)_{ij}}{(\lambda_1-\lambda_2)(\lambda_3-\lambda_2)}\,, \tag{25b}$$

$$X_{i3}X_{j3}^* = \frac{\delta_{ij}\lambda_1\lambda_2 - (\widetilde{H})_{ij}(\lambda_1+\lambda_2) + (\widetilde{H}^2)_{ij}}{(\lambda_1-\lambda_3)(\lambda_2-\lambda_3)}\,. \tag{25c}$$

If $U$ is *unitary*, multiplying the equations above by $U_{\beta i}U_{\alpha j}^*$ and summing over $i$ and $j$, we have

$$\widetilde{U}_{\beta 1}\widetilde{U}_{\alpha 1}^* = \frac{\delta_{\beta\alpha}\lambda_2\lambda_3 - (H)_{\beta\alpha}(\lambda_2+\lambda_3) + \left(H^2\right)_{\beta\alpha}}{(\lambda_2-\lambda_1)(\lambda_3-\lambda_1)}\,, \tag{26a}$$

$$\widetilde{U}_{\beta 2}\widetilde{U}_{\alpha 2}^* = \frac{\delta_{\beta\alpha}\lambda_1\lambda_3 - (H)_{\beta\alpha}(\lambda_1+\lambda_3) + \left(H^2\right)_{\beta\alpha}}{(\lambda_1-\lambda_2)(\lambda_3-\lambda_2)}\,, \tag{26b}$$

$$\widetilde{U}_{\beta 3}\widetilde{U}_{\alpha 3}^* = \frac{\delta_{\beta\alpha}\lambda_1\lambda_2 - (H)_{\beta\alpha}(\lambda_1+\lambda_2) + \left(H^2\right)_{\beta\alpha}}{(\lambda_1-\lambda_3)(\lambda_2-\lambda_3)}\,, \tag{26c}$$

where $H = U\widetilde{H}U^\dagger$ is the Hamiltonian in the flavor basis. One can verify explicitly that the mixing elements are in agreement with the results obtained in refs. [30, 31] for the case of unitary three-flavor neutrino oscillations.

Next let us consider the Jarlskog combinations [43]

$$\widetilde{J}_{\beta\alpha}^{jk} \equiv \mathrm{Im}\left(\widetilde{U}_{\beta j}\widetilde{U}_{\alpha j}^*\widetilde{U}_{\beta k}^*\widetilde{U}_{\alpha k}\right),\qquad \beta\neq\alpha,\quad j\neq k\,, \tag{27}$$

which are antisymmetric in both $jk$ and $\beta\alpha$. For unitary $\widetilde{U}$ or $U$, the Jarlskog combinations (27) are all the same up to an overall sign

$$\widetilde{J}_{\beta\alpha}^{jk} = \frac{\mathrm{Im}\left[\left(H^2\right)_{\alpha\beta}(H)_{\beta\alpha}\right]}{\lambda_{21}\lambda_{31}\lambda_{32}}\sum_l \epsilon_{jkl}\,, \tag{28}$$

where we have defined $\lambda_{jk}\equiv\lambda_j-\lambda_k$ and the totally antisymmetric tensor is defined with $\epsilon_{123}=+1$. From eq. (28), we can verify the following *unitary relations*[5]

$$\widetilde{J}_{\beta\alpha}^{12} + \widetilde{J}_{\beta\alpha}^{13} = \widetilde{J}_{\beta\alpha}^{21} + \widetilde{J}_{\beta\alpha}^{23} = \widetilde{J}_{\beta\alpha}^{31} + \widetilde{J}_{\beta\alpha}^{32} = 0\,, \tag{29}$$

which also hold in the vacuum. Violation of the relations above implies *nonunitarity* in three-flavor neutrino oscillations as we will discuss in Section 3.2.1.

Denoting $H\equiv H_0 + V$ with $H_0\equiv U\Delta U^\dagger$, it follows from eq. (28) that

$$\lambda_{21}\lambda_{31}\lambda_{32}\widetilde{J}_{\beta\alpha}^{jk} = \mathrm{Im}\left[\left(H_0^2 + V^2 + H_0 V + V H_0\right)_{\alpha\beta}(H_0 + V)_{\beta\alpha}\right]\sum_l \epsilon_{jkl}\,. \tag{30}$$

---

[4]If $\mathcal{F}=0$, $\lambda_1=\lambda_2=\lambda_3$ and there is no neutrino oscillation. If $\mathcal{G}=0$, we have a twofold degeneracy $\lambda_1=\lambda_2\neq\lambda_3$ and the system is reduced to a two-flavor scenario in which the solutions are

$$X_{i1}X_{j1}^* + X_{i2}X_{j2}^* = \frac{\delta_{ij}\lambda_3 - (\widetilde{H})_{ij}}{\lambda_3-\lambda_1}\,,\qquad X_{i3}X_{j3}^* = \frac{\delta_{ij}\lambda_1 - (\widetilde{H})_{ij}}{\lambda_1-\lambda_3}\,.$$

[5]For $3+N$ unitary system, the unitary relations are $\sum_k \widetilde{J}_{\beta\alpha}^{jk} = 0$.

If $V$ is diagonal (hence real e.g. the SM matter potential), we have [30, 31, 33]

$$\lambda_{21}\lambda_{31}\lambda_{32}\widetilde{J}^{jk}_{\beta\alpha} = \text{Im}\left\{\left(H_0^2\right)_{\alpha\beta}(H_0)_{\beta\alpha}\right\}\sum_l \epsilon_{jkl} = \Delta_{21}\Delta_{31}\Delta_{32}J^{jk}_{\beta\alpha}, \tag{31}$$

where we have defined $\Delta_{jk} \equiv \Delta_j - \Delta_k$ with $\Delta_j$ defined in eq. (7). The identity above is the Naumov-Harrison-Scott (NHS) identity [44, 45]. A violation of NHS identity implies new physics beyond the three-flavor neutrino oscillation paradigm: nonunitary in $U$ and/or the existence of NSI such that $V$ is not diagonal as we will discuss in Section 3.2.1 and 3.2.2, respectively.

### 3.2.1 Low scale nonunitarity

Here we will discuss the *low scale nonunitary* scenario where sterile neutrinos are light enough to participate in neutrino oscillation but heavy enough such that their fast oscillations can be averaged out [10, 13].[6] In this case, $\widetilde{J}^{jk}_{\beta\alpha}$ defined in eq. (27) is no longer invariant up to an overall sign. Applying eqs. (25a)–(25c) in eq. (27) and without assuming unitary $U$, we obtain

$$\widetilde{J}^{jk}_{\beta\alpha} = \frac{\text{Im}\left\{(U\widetilde{H}^2 U^\dagger)_{\alpha\beta}(U\widetilde{H}U^\dagger)_{\beta\alpha}\right\}}{\lambda_{21}\lambda_{31}\lambda_{32}}\sum_l \epsilon_{jkl}$$
$$+ \sum_l \epsilon_{jkl}\frac{\text{Im}\left\{(UU^\dagger)_{\beta\alpha}\left[(U\widetilde{H}U^\dagger)_{\alpha\beta}\lambda_l^2 - (U\widetilde{H}^2 U^\dagger)_{\alpha\beta}\lambda_l\right]\right\}}{\lambda_{21}\lambda_{31}\lambda_{32}}. \tag{32}$$

As long as $U$ is not unitary, NHS identity does not hold independently of the matter potential. Instead of the unitary relations eq. (29), for nonunitary $U$, we have the new identities

$$\widetilde{J}^{12}_{\beta\alpha} + \widetilde{J}^{13}_{\beta\alpha} = \frac{\text{Im}\left\{(UU^\dagger)_{\beta\alpha}\left[(U\widetilde{H}U^\dagger)_{\alpha\beta}(\lambda_2 + \lambda_3) - (U\widetilde{H}^2 U^\dagger)_{\alpha\beta}\right]\right\}}{\lambda_{12}\lambda_{13}}, \tag{33a}$$

$$\widetilde{J}^{21}_{\beta\alpha} + \widetilde{J}^{23}_{\beta\alpha} = \frac{\text{Im}\left\{(UU^\dagger)_{\beta\alpha}\left[(U\widetilde{H}U^\dagger)_{\alpha\beta}(\lambda_1 + \lambda_3) - (U\widetilde{H}^2 U^\dagger)_{\alpha\beta}\right]\right\}}{\lambda_{21}\lambda_{23}}, \tag{33b}$$

$$\widetilde{J}^{31}_{\beta\alpha} + \widetilde{J}^{32}_{\beta\alpha} = \frac{\text{Im}\left\{(UU^\dagger)_{\beta\alpha}\left[(U\widetilde{H}U^\dagger)_{\alpha\beta}(\lambda_1 + \lambda_2) - (U\widetilde{H}^2 U^\dagger)_{\alpha\beta}\right]\right\}}{\lambda_{31}\lambda_{32}}. \tag{33c}$$

Since $\widetilde{J}^{ij}_{\beta\alpha}$ are antisymmetric in $ij$, the sum of the three terms above vanish[7]

$$\widetilde{J}^{12}_{\beta\alpha} + \widetilde{J}^{13}_{\beta\alpha} + \widetilde{J}^{21}_{\beta\alpha} + \widetilde{J}^{23}_{\beta\alpha} + \widetilde{J}^{31}_{\beta\alpha} + \widetilde{J}^{32}_{\beta\alpha} = 0, \tag{34}$$

and hence we have only two independent combinations. In the vacuum, $\widetilde{H} = \Delta$ and $\lambda_i = \Delta_i$, we obtain the vacuum results of ref. [10]

$$J^{12}_{\beta\alpha} + J^{13}_{\beta\alpha} = -\text{Im}\left\{(UU^\dagger)_{\beta\alpha}U_{\alpha 1}U^*_{\beta 1}\right\}, \tag{35a}$$

$$J^{21}_{\beta\alpha} + J^{23}_{\beta\alpha} = -\text{Im}\left\{(UU^\dagger)_{\beta\alpha}U_{\alpha 2}U^*_{\beta 2}\right\}, \tag{35b}$$

$$J^{31}_{\beta\alpha} + J^{32}_{\beta\alpha} = -\text{Im}\left\{(UU^\dagger)_{\beta\alpha}U_{\alpha 3}U^*_{\beta 3}\right\}. \tag{35c}$$

---

[6]The high scale nonunitary scenario where sterile neutrinos are kinematically forbidden to participate in neutrino oscillation will be explored elsewhere. See also ref. [18] for the study of unitarity violation, with and without kinematically accessible sterile neutrinos.

[7]This follows directly from the definition (27) which gives $\sum_{j,k}\widetilde{J}^{jk}_{\beta\alpha} = 0$.

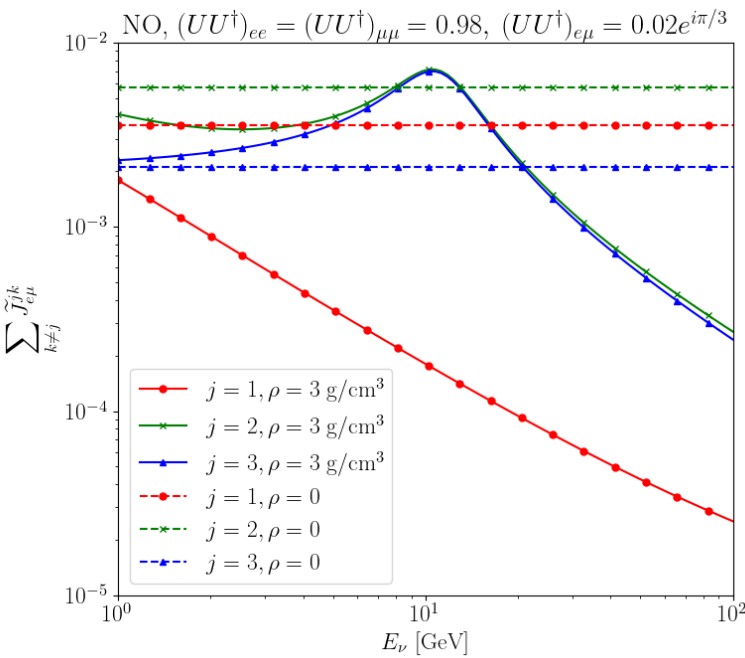

Figure 1: Violation of unitary relations (29) in matter (solid lines) and in vacuum (dashed lines). For the unitary scenario, the quantity plotted here is exactly zero, satisfying the unitary relations (29).

By verifying the relations above in experiments, we can uncover nonunitarity. To illustrate the violation of unitary relations (29), in Figure 1, we plot $\sum_k \widetilde{J}_{\beta\alpha}^{jk}$ for $\left(UU^\dagger\right)_{e\mu} = 0.02 e^{i\pi/3}$ and $\left(UU^\dagger\right)_{ee} = \left(UU^\dagger\right)_{\mu\mu} = 0.98$ for vacuum or constant matter density $\rho = 3$ g/cm$^3$. For the rest of parameters, we have used the global best fit values of Normal mass Ordering (NO) from ref. [46]. In the vacuum, $\sum_k \widetilde{J}_{\beta\alpha}^{jk}$ are constant which sum over $j$ to zero (only the $j = 2$ term represented by green dashed line with crosses is negative). In the matter, $\sum_k \widetilde{J}_{\beta\alpha}^{jk}$ are sensitive to matter potential but still sum over $j$ to zero (only the $j = 2$ term represented by green solid line with crosses is negative). For the unitary scenario, all these quantities are exactly zero as can be seen explicitly in eqs. (33a)–(33c) (in matter) and eqs. (35a)–(35c) (in vacuum) and hence satisfy the unitary relations (29).

In Figure 2, we plot different $jk$ NHS combinations $2E_\nu|\lambda_{21}\lambda_{31}\lambda_{32}\widetilde{J}_{e\mu}^{jk}|^{1/3}$ as a function of neutrino energy $E_\nu$ using eq. (32) for the same parameters as in Figure 1. With the normalization $2E_\nu$, this quantity is a constant in the absence of matter or in unitary scenario with diagonal potential. As a reference, we have plotted the black solid line for the unitary scenario with any diagonal (including zero) matter potential, which is independent of $jk$ and matter density as can be seen explicitly in eq. (31). Due to nonunitarity, all the different combinations $2E_\nu|\lambda_{21}\lambda_{31}\lambda_{32}\widetilde{J}_{e\mu}^{jk}|^{1/3}$ deviate from the black solid line. Furthermore, they are matter-density dependent as opposed to the NHS identity (31).

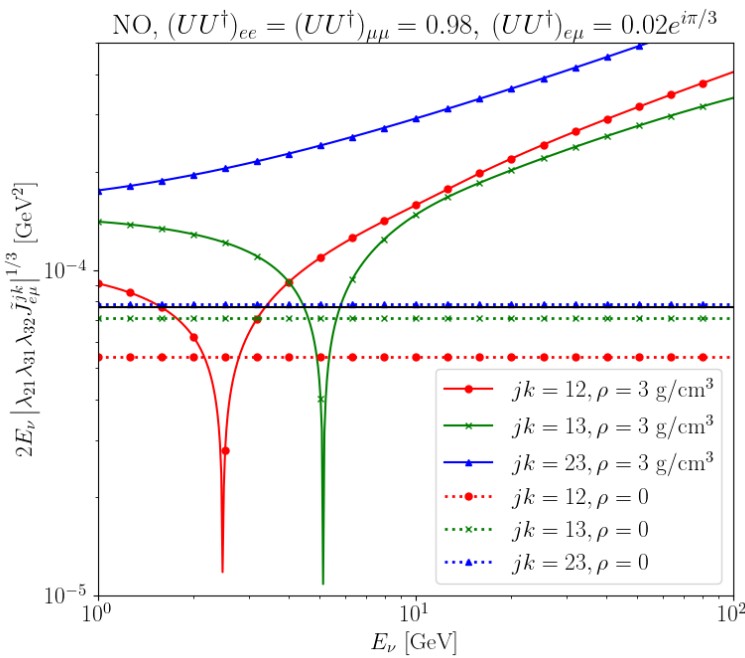

Figure 2: Different combinations $2E_\nu|\lambda_{21}\lambda_{31}\lambda_{32}\widetilde{J}_{e\mu}^{jk}|^{1/3}$ in matter (solid lines) and in vacuum (dotted lines). For reference, the black solid line is for the unitary scenario with standard (or zero) matter potential.

### 3.2.2 Nonstandard neutrino interactions

Due to NSI, the matter potential can be parametrized as [26]

$$V = \sqrt{2}G_F n_e \begin{pmatrix} 1 + \epsilon_{ee} - \frac{1}{2}n_n/n_e & \epsilon_{e\mu} & \epsilon_{e\tau} \\ \epsilon_{e\mu}^* & \epsilon_{\mu\mu} - \frac{1}{2}n_n/n_e & \epsilon_{\mu\tau} \\ \epsilon_{e\tau}^* & \epsilon_{\mu\tau}^* & \epsilon_{\tau\tau} - \frac{1}{2}n_n/n_e \end{pmatrix}, \quad (36)$$

where $G_F$ is the Fermi constant, and $n_e$ and $n_n$ are the number density of electron and neutron, respectively. In the case where $U$ is unitary, from eq. (30), we have a modified NHS identity

$$\lambda_{21}\lambda_{31}\lambda_{32}\widetilde{J}_{\beta\alpha}^{jk} = \Delta_{21}\Delta_{31}\Delta_{32}J_{\beta\alpha}^{jk} + \text{Im}\left\{\sum_{\gamma}\left[(H_0)_{\alpha\gamma}V_{\gamma\beta} + V_{\alpha\gamma}(H_0)_{\gamma\beta}\right](H_0)_{\beta\alpha}\right\}. \quad (37)$$

If the matter potential is diagonal, the original NHS identity (31) is recovered. While it has been suggested in ref. [12] to map nonunitary scenario to NSI scenario and vice-versa, it is important to note that they are *physically distinct*, and give rise to effects which are different qualitatively and quantitatively. If $U$ is unitary, the unitary relations (29) still hold exactly *independently* of the matter potential. In the nonunitary scenario, the unitary relations (29) are violated and instead are replaced by either eqs. (33a)–(33c) in matter or eqs. (35a)–(35c) in vacuum. While the NHS identity never holds for nonunitary scenario, it is violated in the NSI scenario only if the resulting matter potential is *nondiagonal* in which it described by eq. (37).

In Figure 3, we fix $-\epsilon_{ee} = \epsilon_{\mu\mu} = 0.02$ and consider the cases where $\epsilon_{e\mu} = -0.02e^{\mp i\pi/3}$ or $\epsilon_{e\mu} = 0$ while the rest of parameters are set to global best fit values of NO from ref. [46]. The NHS identity (31) is only satisfied when $V$ is diagonal (black solid line) while it is replaced by the new identity (37) when $V$ is nondiagonal (blue dashed and dotted magenta lines). The dip in the magenta dotted line indicates a sign change from negative to positive value. This opens up the new possibility of probing the form of NSI through eq. (37).

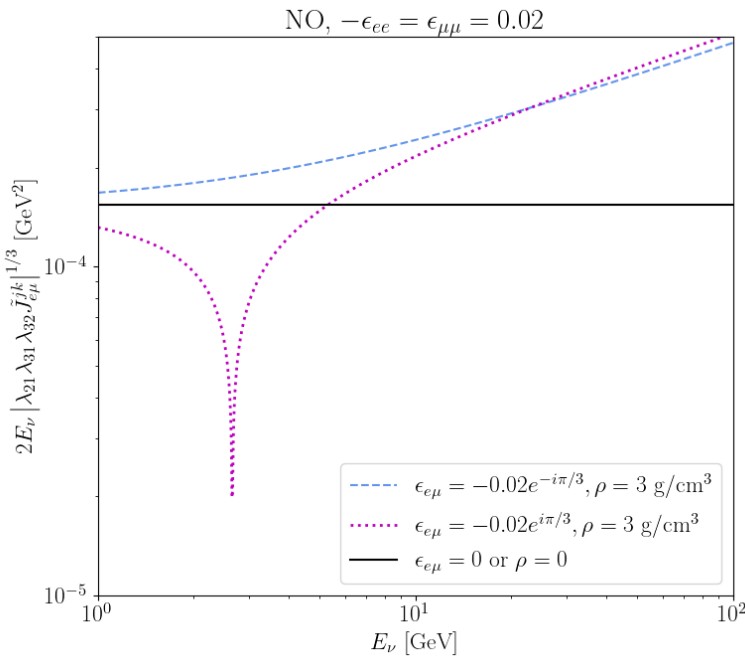

Figure 3: NHS identity (31) is only satisfied in the case where the matter potential is diagonal (black solid line). Otherwise, it is described by eq. (37) which depends on the matter potential (blue dashed and magenta dotted lines).

## 4 Conclusions

In this work, we have derived exact analytic expressions for $(3+N)$-flavor neutrino oscillation probability in arbitrary matter potential in term of Hamiltonian elements and its eigenvalues, which will allow the understanding of the interplay between new physics and neutrino oscillations. In the three-flavor scenario, we have shown that nonunitary scenario is qualitatively and quantitative distinct from NSI scenario. Nonunitary implies violation of unitary relations (29) which are replaced by eqs. (33a)–(33c) in matter or eqs. (35a)–(35c) in vacuum and the NHS identity (31) is also violated. On the other hand, NSI in the unitary scenario preserves (29) while the NHS identity is violated only if the potential is nondiagonal in which case, it is replaced by the new identity (37). In summary, the strategy is to first verify if unitary relations (29) hold. On the one hand, if nonunitarity is discovered, then one would proceed to a more challenging task, but doable in principle, to determine if there is also NSI. On the other hand, if unitary relations (29) hold to a great precision, then one would go on to verify if the matter potential is diagonal or not.

## 5 Acknowledgments

C.S.F. would like to thank Hisakazu Minakata for pointing out the work of Yasuda [34] who was the first to obtain the analytic formula for neutrino oscillation with an arbitrary number of neutrinos.

**Funding information** C.S.F. acknowledges the support by grant 2019/11197-6 and 2022/00404-3 from São Paulo Research Foundation (FAPESP), and grant 301271/2019-4 and 407149/2021-0 from National Council for Scientific and Technological Development (CNPq).

# A  Analytic solutions for $(3+1)$ – flavor scenario

Analytic results for $(3+1)$ were obtained previously in refs. [36–41]. Here we will present the results in a general and compact form. In the $(3+1)$-flavor scenario, the characteristic equation of $\widetilde{H} = \Delta + U^\dagger V U$ is

$$\lambda^4 - \mathcal{T}\lambda^3 + \mathcal{A}\lambda^2 - \mathcal{A}_2\lambda + \mathcal{D} = 0, \tag{A.1}$$

where $\mathcal{T}$, $\mathcal{D}$ and $\mathcal{A}$ are defined in eq. (21) and

$$\mathcal{A}_2 \equiv \frac{1}{6}\left(\mathcal{T}^3 - 3\mathcal{T}\mathcal{T}_2 + 2\mathcal{T}_3\right), \tag{A.2}$$

where $\mathcal{T}_3$ is defined in eq. (22).

The real eigenvalues can be solved using method by Lodovico de Ferrari and are given by

$$\lambda_{1,2} = \frac{\mathcal{T}}{4} - \mathcal{S} \pm \frac{1}{2}\sqrt{2\mathcal{P} - 4\mathcal{S}^2 + \frac{\mathcal{Q}}{\mathcal{S}}}, \qquad \lambda_{3,4} = \frac{\mathcal{T}}{4} + \mathcal{S} \pm \frac{1}{2}\sqrt{2\mathcal{P} - 4\mathcal{S}^2 - \frac{\mathcal{Q}}{\mathcal{S}}}, \tag{A.3}$$

where we have defined

$$\mathcal{P} \equiv \frac{3}{8}\mathcal{T}^2 - \mathcal{A}, \qquad \mathcal{Q} \equiv -\frac{\mathcal{T}^3}{8} + \frac{\mathcal{T}\mathcal{A}}{2} - \mathcal{A}_2, \qquad \mathcal{S} \equiv \frac{1}{2}\sqrt{\frac{2}{3}\mathcal{P} + \frac{2}{3}\mathcal{F}_1\cos\mathcal{G}_1}, \tag{A.4}$$

with

$$\mathcal{F}_1 \equiv \sqrt{\mathcal{A}^2 - 3\mathcal{T}\mathcal{A}_2 + 12\mathcal{D}}, \qquad \mathcal{G}_1 \equiv \frac{1}{3}\arccos\left(\frac{\Delta_1}{2\mathcal{F}_1^3}\right), \tag{A.5}$$

and

$$\Delta_1 \equiv 2\mathcal{A}^3 - 9\mathcal{T}\mathcal{A}\mathcal{A}_2 + 27\mathcal{T}^2\mathcal{D} + 27\mathcal{A}_2^2 - 72\mathcal{A}\mathcal{D}. \tag{A.6}$$

From eq. (18), we have

$$X_{i1}X_{j1}^* = \frac{\delta_{ij}\lambda_2\lambda_3\lambda_4 - (\widetilde{H})_{ij}\left(\lambda_2\lambda_3 + \lambda_2\lambda_4 + \lambda_3\lambda_4\right) + (\widetilde{H}^2)_{ij}\left(\lambda_2 + \lambda_3 + \lambda_4\right) - (\widetilde{H}^3)_{ij}}{(\lambda_2 - \lambda_1)(\lambda_3 - \lambda_1)(\lambda_4 - \lambda_1)}, \tag{A.7a}$$

$$X_{i2}X_{j2}^* = \frac{\delta_{ij}\lambda_1\lambda_3\lambda_4 - (\widetilde{H})_{ij}\left(\lambda_1\lambda_3 + \lambda_1\lambda_4 + \lambda_3\lambda_4\right) + (\widetilde{H}^2)_{ij}\left(\lambda_1 + \lambda_3 + \lambda_4\right) - (\widetilde{H}^3)_{ij}}{(\lambda_1 - \lambda_2)(\lambda_3 - \lambda_2)(\lambda_4 - \lambda_2)}, \tag{A.7b}$$

$$X_{i3}X_{j3}^* = \frac{\delta_{ij}\lambda_1\lambda_2\lambda_4 - (\widetilde{H})_{ij}\left(\lambda_1\lambda_2 + \lambda_1\lambda_4 + \lambda_2\lambda_4\right) + (\widetilde{H}^2)_{ij}\left(\lambda_1 + \lambda_2 + \lambda_4\right) - (\widetilde{H}^3)_{ij}}{(\lambda_1 - \lambda_3)(\lambda_2 - \lambda_3)(\lambda_4 - \lambda_3)}, \tag{A.7c}$$

$$X_{i4}X_{j4}^* = \frac{\delta_{ij}\lambda_1\lambda_2\lambda_3 - (\widetilde{H})_{ij}(\lambda_1\lambda_2 + \lambda_1\lambda_3 + \lambda_2\lambda_3) + (\widetilde{H}^2)_{ij}(\lambda_1 + \lambda_2 + \lambda_3) - (\widetilde{H}^3)_{ij}}{\left(\lambda_1 - \lambda_4\right)\left(\lambda_2 - \lambda_4\right)\left(\lambda_3 - \lambda_4\right)}. \tag{A.7d}$$

Substituting the equations above in eq. (11) (or (16) and (13) for $x$-dependent matter potential), we have the complete analytic solutions for (3+1)-flavor neutrino oscillation probabilities in an arbitrary matter potential. The analytic expression is simple enough to fit into one page and its use to understand $(3+1)$-flavor scenario will be explored elsewhere.

# B  Neutrino oscillation as a probe of new physics with `NuProbe`

We have implemented the analytic expressions derived in this work in a Python code `NuProbe` which is available at https://github.com/shengfong/nuprobe. Out of the box, the code can

deal with up to $(3+4)$-flavor neutrino oscillation system for arbitrary matter potential though the user can readily extend the code to consider beyond $(3+4)$ scenario implementing eq. (18). For $3+N$ system, we parametrize the mixing matrix as $U = U_{\text{NP}}U_0$ where

$$U_{\text{NP}} = R_{3+N-1,3+N}R_{3+N-2,3+N}R_{3+N-3,3+N}...R_{34}R_{24}R_{14}, \tag{B.1}$$

$$U_0 = R_{23}R_{13}R_{12}, \tag{B.2}$$

with $R_{ij}$ the complex rotation matrix in the $ij$-plane which can be obtained from a $(3+N)\times(3+N)$ identity matrix $I$ by replacing the $I_{ii}$ and $I_{jj}$ elements by $\cos\theta_{ij}$, $I_{ij}$ element by $e^{-i\phi_{ij}}\sin\theta_{ij}$, and $I_{ji}$ element by $-e^{i\phi_{ij}}\sin\theta_{ij}$.

Nonunitary three-flavor oscillation can be characterized by three real quantities $\left(UU^\dagger\right)_{\alpha\alpha} \neq 1$ and three complex quantities $\left(UU^\dagger\right)_{\alpha\beta} \neq 0$ with $\alpha \neq \beta$. To parametrize them, we will go through $\alpha$-parametrization by first considering $U = \alpha U_0$ where we have chosen $\alpha$ to be a lower triangle matrix with real diagonal and complex off-diagonal entries

$$\alpha = \begin{pmatrix} \alpha_{11} & 0 & 0 \\ \alpha_{21} & \alpha_{22} & 0 \\ \alpha_{31} & \alpha_{31} & \alpha_{33} \end{pmatrix}. \tag{B.3}$$

Then, we solve for

$$\begin{aligned}
\alpha_{11} &= \sqrt{(UU^\dagger)_{ee}}, \\
\alpha_{21} &= \frac{\left(UU^\dagger\right)^*_{e\mu}}{\sqrt{(UU^\dagger)_{ee}}}, \\
\alpha_{22} &= \sqrt{(UU^\dagger)_{\mu\mu} - \frac{\left|(UU^\dagger)_{e\mu}\right|^2}{(UU^\dagger)_{ee}}}, \\
\alpha_{31} &= \frac{\left(UU^\dagger\right)^*_{e\tau}}{\sqrt{(UU^\dagger)_{ee}}}, \\
\alpha_{32} &= \frac{1}{\alpha_{22}}\left[\left(UU^\dagger\right)^*_{\mu\tau} - \frac{\left(UU^\dagger\right)_{e\mu}\left(UU^\dagger\right)^*_{e\tau}}{(UU^\dagger)_{ee}}\right], \\
\alpha_{33} &= \sqrt{(UU^\dagger)_{\tau\tau} - \frac{\left|(UU^\dagger)_{e\tau}\right|^2}{(UU^\dagger)_{ee}} - |\alpha_{32}|^2}.
\end{aligned} \tag{B.4}$$

Written in this way, the inputs are $U_0$ and $\left(UU^\dagger\right)_{\alpha\beta}$ which are independent of parametrization.

For the SM neutrinos traveling through an electrically neutral matter consisting of number density $n_e$ of electron and $n_n$ of neutron, we have

$$V = \sqrt{2}G_F \begin{pmatrix} n_e - \frac{1}{2}n_n & 0 & 0 \\ 0 & -\frac{1}{2}n_n & 0 \\ 0 & 0 & -\frac{1}{2}n_n \end{pmatrix}, \tag{B.5}$$

where $G_F$ is the Fermi constant, $n_e = N_A Y_e \rho$ and $n_n = Y_n n_e$ where $N_A = 6.02214076\times10^{23}/\text{mol}$ is the Avogadro constant, $Y_e$ the average number of electrons per nucleon, $Y_n$ the average number of neutrons per electron, and the matter density $\rho$ is given in unit of g/cm$^3$. For Earth-crossing neutrinos, we implement the simplified (Preliminary Reference Earth Model)

PREM model [47] with $\rho$ as a function of the distance from the center of the Earth $r$ as

$$\rho = \begin{cases} 13, & 0 < r < 0.19R_e, \\ 11, & 0.19R_e < r < 0.55R_e, \\ 5, & 0.55R_e < r < 0.90R_e, \\ 3.5, & 0.90R_e < r < R_e, \end{cases} \tag{B.6}$$

where $R_e = 6371$ km is the Earth's radius. Modification due to NSI can be specified in the program.

For the $(3+3)$-flavor scenario where neutrinos are quasi-Dirac, it is convenient to parametrize the $6 \times 6$ unitary matrix as [21, 22]

$$U = \frac{1}{\sqrt{2}} \begin{pmatrix} AU_0 + B & i(AU_0 - B) \\ CU_0 + D & i(CU_0 - D) \end{pmatrix}, \tag{B.7}$$

where $U_0$ is a $3 \times 3$ unitary matrix while the rest of $3 \times 3$ matrices are constrained by $UU^\dagger = U^\dagger U = I_{3\times3}$. In the Dirac limit, $A = I_{3\times3}$, $B = C = 0$ and $D = V_0$ is an arbitrary unitary matrix. An explicit Euler parametrization can be carried out as follows

$$U = U_{\text{NP}} U_0 Y, \tag{B.8}$$

where $U_{\text{NP}}$ and $U_0$ are given by eqs. (B.1) and (B.2), respectively, and

$$Y \equiv \frac{1}{\sqrt{2}} \begin{pmatrix} I_{3\times3} & iI_{3\times3} \\ I_{3\times3} & -iI_{3\times3} \end{pmatrix}. \tag{B.9}$$

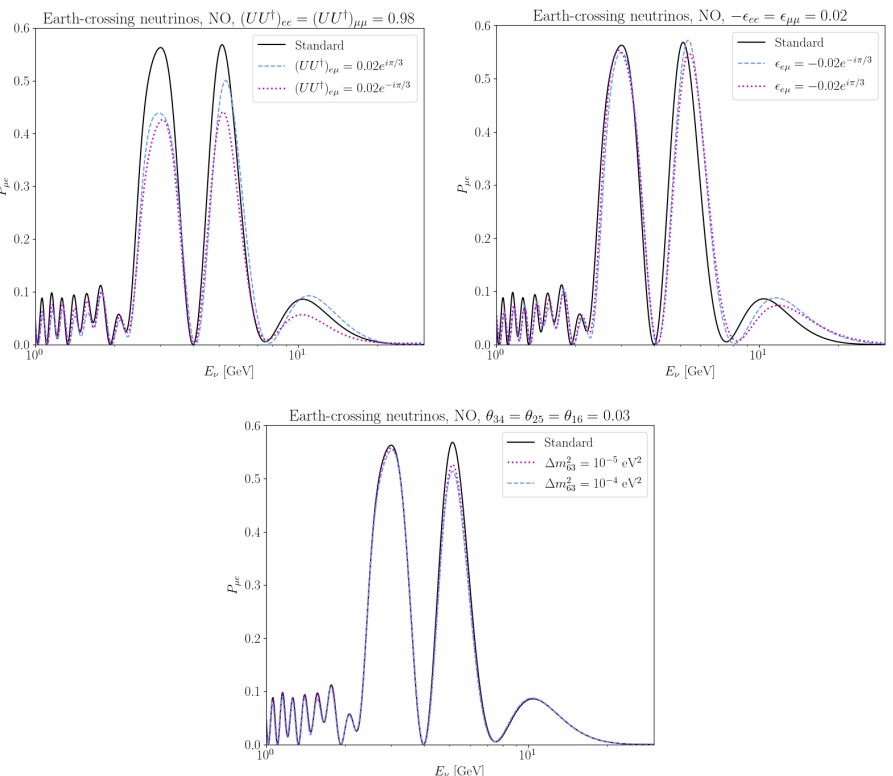

Figure 4: Probability of $\nu_\mu \to \nu_e$ using the simplified PREM model (B.6) for the nonunitary, NSI and quasi-Dirac neutrino scenarios.

In Figure 4, we show the examples of Earth-crossing neutrinos using the simplified PREM model (B.6) for nonunitary, NSI and quasi-Dirac neutrino scenarios. For the rest of parameters, we have set them to the global best fit values of NO from ref. [46]. The codes to generate all the plots in this work can be obtained from https://github.com/shengfong/nuprobe.

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
