# Peer review of "Analytic Neutrino Oscillation Probabilities"

_SciPost Physics, doi:SciPost Phys. 15, 013 (2023)_

## Round 2 · Referee Report · Anonymous (Referee 1) · 2023-1-17

Strengths
- Clearly written
- Has utility for the community
Weaknesses
- Certain points could be clarified as detailed in the report
Report
This paper derives analytic formula for neutrino oscillations in the (3+N)flavour scenario. It also considers the differences between the NSI and non-unitary PMNS (i.e. 3+N flavour) scenarios. It would be helpful if the author could clarify:
-
The author states that the two scenarios are qualitatively and quantitively different. Could this be demonstrated in a plot? The two figures show the different NHS and unitary behaviours of each scenario separately i.e. what would we expect to measure for the NHS/Jarlskog identities.
-
The paper discusses how the non-unitary / non-diagonal NSI scenarios can be differentiated, however using this method it does not seem possible to distinguish between a diagonal or non-diagonal potential in the non-unitary scenario. Is there any possible resolution to this problem? How does the NHS identity behave in this in this scenario?
-
How is the NHS identity affected by non-unitarity? It may be helpful to the reader if the author produces a figure similar to Fig 1 but for the NHS identity in the non-unitary scenario.
Author: Chee Sheng Fong on 2023-01-20 [id 3254]
(in reply to Report 1 on 2023-01-17)I would like to thank the referee for highlighting the important points of differentiating between nonunitary and NSI scenarios.
1) In the unitary case, the quantities in Figure 1 will be exactly zero since the unitary relations (29) will hold. And the NHS identity (31) will also be satisfied if the potential remains diagonal.
2) This is an important and difficult question! A reasonable strategy is to first verify if unitary relations (29) hold. On the one hand, if nonunitarity is discovered, then one would proceed to a more challenging task, but doable in principle, to determine if there is also NSI. For example, in the general eq. (32), on top of nonunitary parameters, if further NSI parameters are need in the Hamiltonian. On the other hand, if unitary relations (29) hold to a great precision, then one could go on to verify if the matter potential is diagonal or not.
3) In the attached figure, the NHS combinations for the case of nonunitarity are shown. The solid black thin line is the reference line for the unitary case with any diagonal potential (including zero). It is true that this can be confused with the case of NSI (Figure 2), especially if there are both nonunitary and NSI. So, the key is really to first identify if unitary relations hold as mentioned in the previous point and then go on from there.
Attachment:

---

## Round 3 · Referee Report · Anonymous · 2023-1-30

Strengths

1. clearly written useful result

Weaknesses

1. I believe the author addressed the issues we found with the paper

Report

Yes, the author's response is complete and careful, the manuscript is ready for publication

---

## Round 3 · Author Response

The article has been revised to take into account the queries of the referee.

---

## Round 3 · List of Changes

1) A sentence is added in the text as well as in the caption of Figure 1 to stress that the quantity plotted is exactly zero in the unitary scenario.

2) A new Figure 2 with a paragraph added in end of Section IIIB1 to show how the NHS combinations are modified in the presence of nonunitarity.

3) Normalization in Figure 3 (the previous Figure 2) is changed for easier comparison with Figure 2.

4) A discussion on the strategy to distinguish nonunitary and NSI scenarios is added at the end of the conclusions.

5) A new reference [34] of Yasuda is added. He was the first to obtain analytical oscillation probability for arbitrary number of neutrinos.

---

## Editorial Decision

published